# Hereditary Congenital Methemoglobinemia Diagnosed at the Age of 79 Years: A Case Report

**DOI:** 10.3390/medicina59030615

**Published:** 2023-03-20

**Authors:** Marohito Nakata, Naoko Yokota, Kazuhiko Tabata, Takuya Morikawa, Hiroki Shibata, Tsuneaki Kenzaka

**Affiliations:** 1Department of Cardiology, Urasoe General Hospital, Urasoe 901-2132, Japan; xbqhf498@yahoo.co.jp (M.N.); naokoyixx86@yahoo.co.jp (N.Y.); 2Department of Internal Medicine, Naha City Hospital, Naha 902-8511, Japan; 39hiko-naha@nch.naha.okinawa.jp; 3Division of Genomics, Medical Institute of Bioregulation, Kyushu University, Fukuoka 812-8582, Japan; morikawa@gen.kyushu-u.ac.jp (T.M.); hshibata@gen.kyushu-u.ac.jp (H.S.); 4Division of Community Medicine and Career Development, Kobe University Graduate School of Medicine, Kobe 652-0032, Japan

**Keywords:** cyanosis, hereditary congenital methemoglobinemia, old age

## Abstract

*Background*: Cardiopulmonary disorders are the most common cause of central cyanosis, and methemoglobinemia is often overlooked in the differential diagnosis of patients with central cyanosis. In most cases, methemoglobinemia is acquired and hereditary congenital methemoglobinemia is rare. Only a few case reports of congenital methemoglobinemia can be found in PubMed. To date, only four cases of congenital methemoglobinemia diagnosed after the age of 50 years have been reported. *Case Presentation*: A 79-year-old Japanese woman presented at our hospital with the chief complaints of dyspnea and cyanosis. She exhibited cyanosis of the lips and extremities, and her SpO_2_ was 80%, with oxygen administration at 5 L/min. Blood gas analysis revealed a PaO_2_ of 325.4 mmHg and methemoglobin level of 36.9%. The SpO_2_ and PaO_2_ values were dissociated, and methemoglobin levels were markedly elevated. Genetic analysis revealed a nonsynonymous variant in the gene encoding nicotinamide adenine dinucleotide cytochrome (NADH) B5 reductase 3 (*CYB5R3*), and the patient was diagnosed with congenital methemoglobinemia. *Conclusions*: It is important to consider methemoglobinemia in the differential diagnosis of patients with central cyanosis. At 79 years of age, our patient represents the oldest patient with this diagnosis. This report indicates that it is crucial to consider the possibility of methemoglobinemia regardless of the patient’s age.

## 1. Introduction

Methemoglobinemia is a disease that can cause cyanosis and shortness of breath. Pulse oximeters generally rely on data from healthy individuals with low levels of carboxyhemoglobin and methemoglobin. Hence, pulse oximeter signals may be invalid in patients with underlying conditions. Dissociation between SpO_2_ and PaO_2_ values may aid in the diagnosis of methemoglobinemia. Methemoglobinemia can be either congenital or acquired, and the most common cause of congenital methemoglobinemia is functional variants in the gene encoding NADH-cytochrome B5 reductase 3 (*CYB5R3*). Methemoglobin is a form of the protein hemoglobin in which the iron is in the ferric state, rather than the normal ferrous state. Although a small percentage of methemoglobin is present in healthy individuals, an increase in methemoglobin content occurs due to loss of function of *CYB5R3*. Methemoglobinemia is caused by functional variants of *CYB5R3* transmitted in an autosomal recessive manner. Herein, we present a case of congenital methemoglobinemia diagnosed at the age of 79 years; this may be the oldest patient with this diagnosis reported to date.

## 2. Case Report

### 2.1. Case Presentation

A 79-year-old Japanese woman presented to our hospital with the chief complaints of dyspnea and cyanosis. She had a history of pulmonary tuberculosis, lung cancer (following partial right lobe resection and radiotherapy), and bronchiectasis. In addition, the patient had undergone home oxygen therapy 6 months prior due to chronic respiratory failure. Two hours before arriving at our hospital, the patient reported dyspnea and her usual oxygen dosage of 2 L/min was increased to 5 L/min. Nonetheless, her dyspnea did not improve, and her SpO_2_ decreased to as low as 80%. This prompted her to seek emergency care at our hospital.

### 2.2. Investigation

At the time of examination, the patient’s vital signs were as follows: blood pressure, 134/62 mmHg; pulse, 60 beats per minute; respiratory rate, 23 breaths per minute, SpO_2_, 80% (while receiving 5 L/min of oxygen via a mask); and body temperature, 37.3 °C. Physical examination revealed cyanosis of the lips and extremities, and auscultation revealed diminished breath sounds (but no wheezes or crackles). Chest radiography revealed no infiltrative shadows.

### 2.3. Differential Diagnosis

Despite the absence of infiltrative shadows on chest radiography, the patient was admitted to our hospital with a diagnosis of acute bronchitis based on her history of bronchiectasis and the presence of low-grade fever and purulent sputum. Gram staining of the sputum revealed the presence of Gram-negative bacilli. Because *Pseudomonas aeruginosa* had been detected previously in sputum culture tests, antimicrobial therapy with ceftazidime (1 g administered every 12 h) was initiated. Despite the administration of oxygen, the patient’s SpO_2_ remained low. Therefore, we initiated noninvasive positive pressure ventilation, which also did not improve the SpO_2_ value. Arterial blood gas analysis revealed a high PaO_2_ (325.4 mmHg)—indicating a marked discrepancy with the SpO_2_ of 80%—and elevated methemoglobin (36.9%; Table 1). Based on these findings, the patient was diagnosed with methemoglobinemia.

### 2.4. Genetic Analysis

Genetic testing was performed since the chronic elevation of methemoglobin levels strongly suggested the possibility of congenital methemoglobinemia. We extracted genomic DNA from the patient with approval from the Ethics Committee of the School of Medicine, Kyushu University (#680-01). Since *CYB5R3* is known to be the most common responsible gene in congenital methemoglobinemia, we examined all exons of *CYB5R3* using Sanger sequencing. We observed a rare homozygous nonsynonymous variant in exon 3 (NM_000398.7:c.173G>A [p.Arg58Gln], rs121965007; Figure 1) [1]. The nucleotide variant has a CADD score of 27.5 (https://cadd.gs.washington.edu, accessed on 31 January 2023) and has been previously reported as a causative variant in only three Japanese patients with methemoglobinemia [2]. We did not observe any other sequence variants in the other eight exons examined in *CYB5R3*. The resulting amino acid substitution, Arg58Gln, was predicted to be probably damaging by PolyPhen2 (http://genetics.bwh.harvard.edu/pph2/, accessed on 31 January 2023) and damaging by SIFT (https://sift.bii.a-star.edu.sg, accessed on 31 January 2023) software. Since the variant is located in the flavin adenine dinucleotide (FAD)-binding domain, it is likely to affect binding affinity to FAD, resulting in the reduced efficiency of electron transfer (Figure 2) [3,4].

## 3. Outcome and Follow-Up

Following the diagnosis of methemoglobinemia and administration of 8 mL methylene blue (0.2 mL/kg), the SpO_2_ increased to 99% within approximately 10 min (Figure 3 and Figure 4), with resolution of cyanosis (Figure 5). The color of the blood samples changed from dark red to red (Figure 6), and the methemoglobin level decreased to 2.0%. The patient had previously visited our hospital, and a review of her medical records revealed that her methemoglobin level had been elevated for some time (10% and 20% at 5 and 2 years prior to the latest visit, respectively). Although we considered the possibility of acquired methemoglobinemia, there was no history of drug use that would potentially induce methemoglobinemia. Genetic testing revealed congenital methemoglobinemia. After starting ascorbic acid therapy (750 mg/day), the methemoglobin level remained in the range of 6–8% and SpO_2_ remained above 90%, allowing for the discontinuation of home oxygen therapy. In this instance, the presence of *Pseudomonas aeruginosa* was confirmed through sputum culture, and the patient received a 5-day regimen of ceftazidime.

## 4. Discussion

### 4.1. First and Second Novelty

Cardiopulmonary diseases are the most common cause of central cyanosis, and methemoglobinemia is often overlooked in the differential diagnosis [5]. However, methemoglobinemia should definitely be considered when there is a discrepancy between SpO_2_ and PaO_2_ values [5,6]. To the best of our knowledge, only four cases of congenital methemoglobinemia diagnosed after the age of 50 years have been reported to date [5,6,7,8,9]. As such, our patient is the oldest person (79 years old) in whom the diagnosis of congenital methemoglobinemia has been made.

### 4.2. Significance of the First and Second Novelty

Central cyanosis is induced by hypoxemia and is commonly caused by cardiopulmonary diseases. The specific causes of central cyanosis include right–left shunting (leading to atrial septal defects, transposition of the great arteries, tetralogy of Fallot, or pneumonia) and mismatched ventilation and blood flow (leading to bronchial asthma, emphysema, atelectasis, or pulmonary edema) [10]. However, methemoglobinemia is often overlooked in the differential diagnosis of patients with central cyanosis [5]. Our patient had a history of pulmonary tuberculosis, lung cancer, and bronchiectasis and had been on home oxygen therapy for 6 months. Therefore, the cause of cyanosis was initially thought to be pulmonary disease. However, the SpO_2_ did not increase, and cyanosis did not improve with noninvasive positive pressure ventilation. In addition, we noticed a dissociation between the SpO_2_ and PaO_2_ values and observed an increase in the level of methemoglobin.

Pulse oximeters do not directly measure oxygen saturation in the blood. Instead, the measurements are obtained by transmitting two wavelengths of light (red light, 660 nm; infrared light, 940 nm) through tissue (usually a finger or an earlobe) and detecting light that is not attenuated by the tissue bed [10]. Because SpO_2_ is based on data from healthy individuals with low levels of carboxyhemoglobin and methemoglobin, the values obtained from pulse oximetry may be invalid for patients with hemoglobin with different absorbance spectra [11]. The absorption coefficient of methemoglobin at 660 nm is almost the same as that at 940 nm, resulting in a 1:1 ratio of red light to infrared light. The SpO_2_ value is approximately 85% for this ratio, and SpO_2_ measures tends to approach 85% with an increase in the concentration of methemoglobin. An animal study reported that when methemoglobin concentration reaches 30–35%, with SpO_2_ plateaus between 82–86%; thereafter, SpO_2_ values become independent of methemoglobin concentration [11]. Our patient showed a methemoglobin concentration of 36.9% and SpO_2_ of 81%. This is largely consistent with the results of the previous animal study. In methemoglobinemia, cyanosis depends not on the concentration of methemoglobin, but on the total amount of methemoglobin present (calculated as hemoglobin value × %methemoglobin). Cyanosis occurs when the total amount of methemoglobin exceeds 1.5 g/dL. Therefore, a high erythrocyte count contributes to cyanosis, and anemic conditions cause cyanosis to be less prominent [12].

Congenital methemoglobinemia has three genetic causes [6,7]. The most common cause is *CYB5R3* deficiency, followed by hemoglobin M disease and cytochrome B5 deficiency. There are two types of *CYB5R3* deficiencies: type I deficiency is limited to erythrocytes, whereas type II deficiency occurs in all tissues. Patients with type I disease exhibit mild symptoms and have a normal lifespan, whereas those with type II disease exhibit cyanosis, experience neurological damage, and have a significantly shorter lifespan [7]. Type I patients are usually asymptomatic, although they may exhibit mild cyanosis in childhood. However, the decline of cardiopulmonary function with aging may cause additional symptoms, such as shortness of breath and fatigue. In the present case, genetic analysis (performed at the Division of Genomics, Medical Institute of Bioregulation, Kyushu University) revealed a mutation in the *CYB5R3* gene—specifically, an arginine-to-glutamine substitution—that is known to cause type I disease [1]. Because the patient had type I disease, she was able to survive until the age of 79 years without major symptoms. Congenital methemoglobinemia due to *CYB5R3* deficiency is inherited in an autosomal recessive manner and has been reported in certain populations [6,13,14,15]. Although there were no diagnosed cases of congenital methemoglobinemia in the family history of our patient, 28 cases of congenital methemoglobinemia have been reported in 12 families from the patient’s native place [16]. Kiyama et al. have previously reported 15 cases of congenital methemoglobinemia [17] and found that congenital methemoglobinemia is common on the islands of Henza and Miyagi in Okinawa, Japan. They surveyed 2876 islanders on the islands of Henza and Miyagi and found an additional 13 cases, leading to a total of 28 cases in 12 families. Of the 28 cases, 46.4% (13 of 28) were male; mean age was 27.3 years (median: 22, standard deviation: 15.8), 3 were over the age of 50 years, and 1 was 70-year-old; 25 were asymptomatic, 1 had epilepsy, and 2 had an intellectual disability. These findings suggest that the local population at this place may have a higher prevalence of congenital methemoglobinemia. We informed the patient’s daughter of this possibility.

### 4.3. Clinical Implications

Methemoglobinemia should be considered in the differential diagnosis of patients with central cyanosis, especially when there is significant dissociation between SpO_2_ and PaO_2_ values. This difference is generally greater than 5% in cases of methemoglobinemia [5]. A methemoglobin level of 20% suggests enzyme deficiency [5]. There are few subjective symptoms in the case of type I disease. However, the life expectancy of the patient is not shortened, as exemplified by the present case in which the diagnosis was not made until the age of 79 years.

## 5. Conclusions

Herein, we present a case of congenital methemoglobinemia first diagnosed at the age of 79 years. Cyanosis is the most common symptom indicative of congenital methemoglobinemia [5,6,7,8,9]. When evaluating central cyanosis, it is important to check for any dissociation between SpO_2_ and PaO_2_ values; if the difference exceeds 5%, methemoglobinemia should be considered as a possible diagnosis. Most cases of methemoglobinemia are acquired, and patients with type I *CYB5R3* deficiency can have a prolonged lifespan. Furthermore, these patients may not be diagnosed until later in life, as observed in the present case.

## Figures and Tables

**Figure 1 medicina-59-00615-f001:**
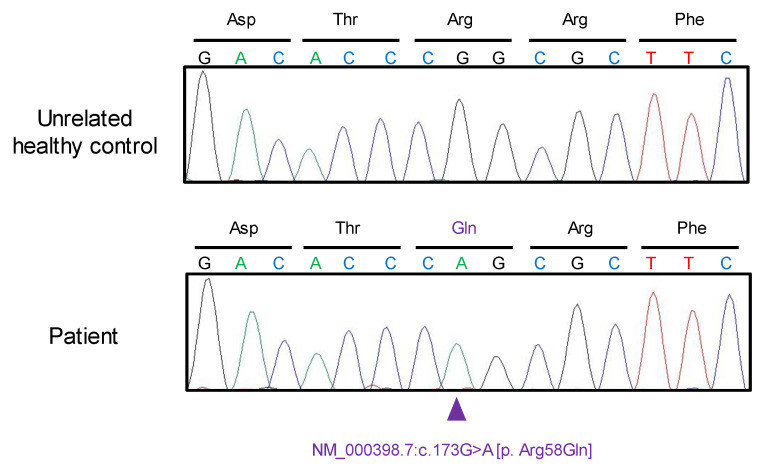
Electropherogram of the region of the variant NM_000398.7:c.173G>A [p.Arg58Gln]. The location of the variant is shown using a purple arrowhead. Abbreviations: Asp: Aspartic Acid; Thr: Phenylalanine; Arg: Arginine; Phe: Phenylalanine; Gln: Glutamine.

**Figure 2 medicina-59-00615-f002:**
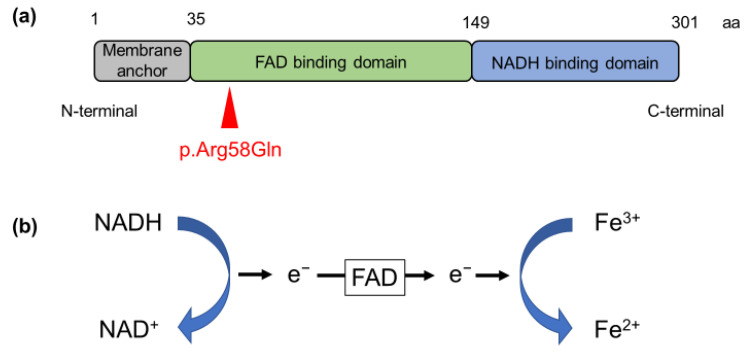
Enzymatic activity of CYB5R3. (**a**) Structure of CYB5R3 protein. CYB5R3 is composed of three domains. The variant we identified is shown by a red arrowhead. (**b**) The process by which iron (Fe) ions are reduced by CYB5R3. Flavin adenine dinucleotide (FAD) is responsible for the transfer of electrons. Abbreviations: aa: amino acid; NADH: nicotinamide adenine dinucleotide cytochrome; NAD: nicotinamide adenine dinucleotide.

**Figure 3 medicina-59-00615-f003:**
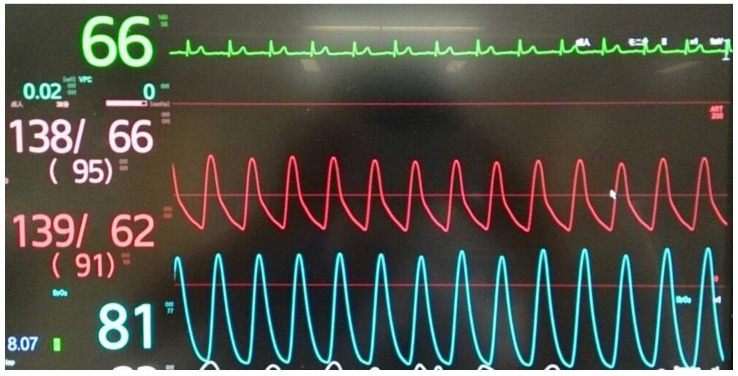
SpO_2_ values and other vital signs before the administration of methylene blue.

**Figure 4 medicina-59-00615-f004:**
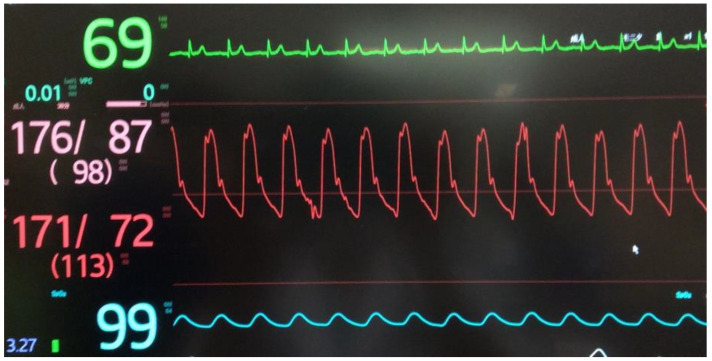
SpO_2_ values and other vital signs after the administration of methylene blue.

**Figure 5 medicina-59-00615-f005:**
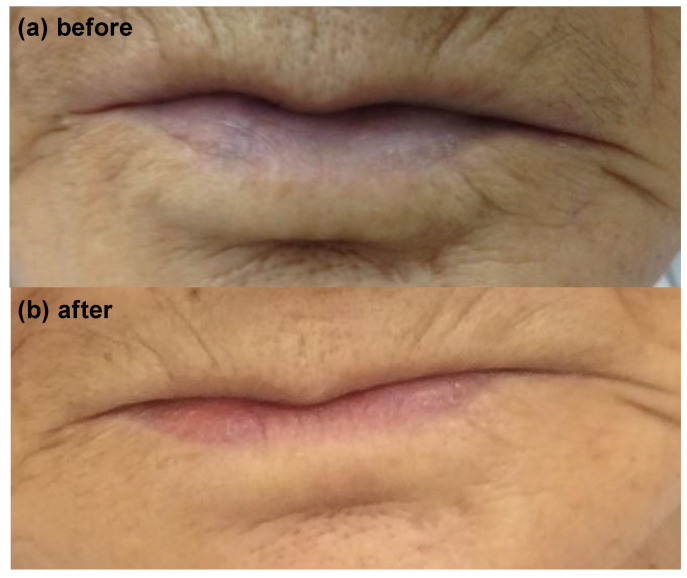
Lips of the patient before (**top**, showing cyanosis) and after (**bottom**, no cyanosis) the administration of methylene blue.

**Figure 6 medicina-59-00615-f006:**
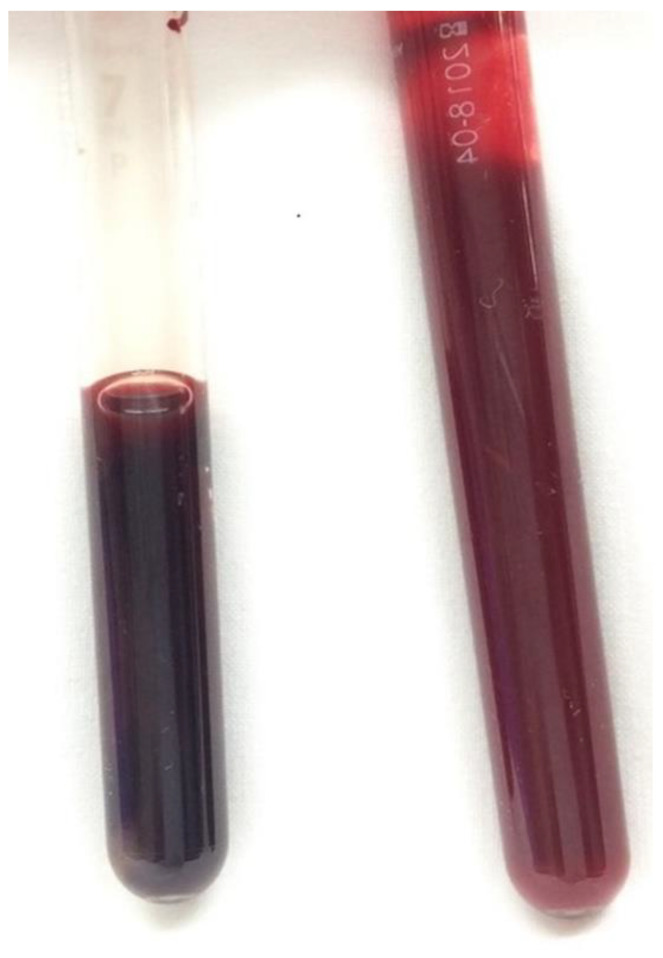
Color of the patient’s blood before (**left**) and after (**right**) the administration of methylene blue.

**Table 1 medicina-59-00615-t001:** Laboratory test results at admission.

Parameter	Recorded Value	Standard Value
White blood cell count	5800/µL	3300–8600/µL
Hemoglobin	13.2 g/dL	11.5–15.0 g/dL
Platelet count	20.9 × 10^4^/µL	15–35 × 10^3^/µL
C-reactive protein	1.31 mg/dL	≤0.14 mg/dL
Total protein	7.3 g/dL	6.6–8.1 g/dL
Albumin	3.7 g/dL	4.1–5.1 g/dL
Aspartate aminotransferase	13 U/L	13–30 U/L
Alanine aminotransferase	8 U/L	7–23 U/L
Lactase dehydrogenase	131 U/L	124–222 U/L
Blood urea nitrogen	14.0 mg/dL	8–20 mg/dL
Creatinine	0.93 mg/dL	0.46–0.79 mg/dL
Sodium	139 mEq/L	138–145 mEq/L
Potassium	4.1 mEq/L	3.6–4.8 mEq/L
Chloride	104 mEq/L	101–108 mEq/L
Glucose	100 mg/dL	75–110 mg/dL
Atrial blood gas		
pH	7.457	7.35–7.45
pCO_2_	36.8 mmHg	35–45 mmHg
pO_2_	325.4 mmHg	80–90 mmHg
O_2_ saturation	99.8%	92.0–98.5%
HCO_3_	25.4 mmol/L	21–30 mmol/L
Lactate	1.10 mmol/L	0.5–1.6 mmol/L
Methemoglobin	36.9%	0–1.5%

Abbreviations: pCO_2_: partial pressure of arterial carbon dioxide; pO_2_:partial pressure of arterial oxygen; HCO_3_: bicarbonate.

## Data Availability

All data generated or analyzed during this study are included in this published article.

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
