# Peer review of "Hereditary Congenital Methemoglobinemia Diagnosed at the Age of 79 Years: A Case Report"

_medicina, 2023, doi:10.3390/medicina59030615_

Round 1

Reviewer 1 Report

The authors report having diagnosed congenital methemoglobinemia in the oldest person to date, namely a 79-year old female patient of Japanese ethnicity.  The patient’s methemoglobinemia, due to a point mutation (arginine to glutamine) in the Cyb5R3 gene, was found to only affect erythrocytes (type 1 methemoglobinemia).  The authors report the patient as having a clinical history of lung cancer, pulmonary tuberculosis and bronchiectasis, but no previous diagnosis of congenital methemoglobinemia despite requiring home oxygen therapy (2-5L/min) for chronic respiratory failure in the previous 6 months.  The authors speculate that the patient’s old age, facilitated by the type I phenotype, may underlie the previously missed diagnosis of methemoglobinemia in the patient. The manuscript is well written and very interesting.  There are some edits, clarifications, and additions, as indicated below, that would improve the manuscript for publication.

1) Abstract: The authors state that “to date, only 4 cases of congenital methemoglobinemia diagnosed after the age of 50 have been documented.” The substance of this statement appears twice in the abstract (lines 17-18, 26-27).

2) Table 1: Given the patient’s high LDH level (385U/L), was any attempt made to determine whether a hemolytic state was also present?  Does type I methemoglobinemia contribute to RBC hemolysis? What were RBC and hematocrit levels in the patient? 

3) Line 86: The link for SIFT does not work and looks like it may have undergone updating after the manuscript was submitted for publication.

4) Lines 148-150: The patient showed an SpO2 of 81% with a methemoglobin concentration of 36.9%, which the authors indicate is not in accordance with the literature, wherein animal studies have shown that SpO2 reaches a plateau of 85% when methemoglobin concentration exceeds 30%. To explain this disparity, the authors contend that “absorbance values change with a decrease in hemoglobin level, which may explain why the patient’s SpO2 was lower than 85% due to a potentially invalid pulse oximetry signal (line 35).”  This explanation seems inadequate given the patient’s hemoglobin level of 13.2 g/dL, which is within the normal range.  

5) Lines 171-172: “28 cases of congenital methemoglobinemia have been reported in 12 families from the patient’s place of origin.” Contrasted with only 4 known reports of congenital methemoglobinemia being diagnosed after the age of 50 (lines 17-18, line 121), the manuscript would be significantly improved by inclusion of information on these other 28 cases (age of diagnosis, gender, clinical history). 

6) Introduction: More background on how Cyb5R3 functions in vivo and how it affects red blood cells (for instance, reduction of ferric iron to ferrous iron to enable oxygen transport) would be useful to the reader.  How the Arg58Gln point mutation affects Cyb5R3 function, if known, should be included.

7) Reference 10:  incomplete citation. See https://lib.dmu.edu/db/uptodate/cite.

Author Response

Response: Thank you for taking the time and effort to review our paper.

We appreciate your encouraging comments.

  • Abstract: The authors state that “to date, only 4 cases of congenital methemoglobinemia diagnosed after the age of 50 have been documented.” The substance of this statement appears twice in the abstract (lines 17-18, 26-27)

Response: Thank you for pointing this out. We have deleted this sentence from the abstract.

  • Table 1: Given the patient’s high LDH level (385U/L), was any attempt made to determine whether a hemolytic state was also present?  Does type I methemoglobinemia contribute to RBC hemolysis? What were RBC and hematocrit levels in the patient? 

Response: Thank you for pointing this out. There was a typographical error in the manuscript. LDH level was 131 U/L, alkaline phosphatase (ALP) level was 385 U/L, RBC count was 408×104/μL, and hematocrit was 38.7%. As LDH level was within the normal range, we did not provide RBC count and hematocrit value in the text.

  • Line 86: The link for SIFT does not work and looks like it may have undergone updating after the manuscript was submitted for publication.

Response: Thank you for pointing this out. We have corrected this and inserted a valid link.(Lines 86-89)

  • Lines 148-150: The patient showed an SpO2 of 81% with a methemoglobin concentration of 36.9%, which the authors indicate is not in accordance with the literature, wherein animal studies have shown that SpO2 reaches a plateau of 85% when methemoglobin concentration exceeds 30%. To explain this disparity, the authors contend that “absorbance values change with a decrease in hemoglobin level, which may explain why the patient’s SpO2 was lower than 85% due to a potentially invalid pulse oximetry signal (line 35).”  This explanation seems inadequate given the patient’s hemoglobin level of 13.2 g/dL, which is within the normal range.  

Response: Thank you for this valuable suggestion. We read the reference again and revised the manuscript based on your suggestion.

An animal study reported when methemoglobin concentration reaches 30–35%, SpO2 plateaus between 82–86%; thereafter, SpO2 values become independent of methemoglobin concentration [11].

  • Lines 171-172: “28 cases of congenital methemoglobinemia have been reported in 12 families from the patient’s place of origin.” Contrasted with only 4 known reports of congenital methemoglobinemia being diagnosed after the age of 50 (lines 17-18, line 121), the manuscript would be significantly improved by inclusion of information on these other 28 cases (age of diagnosis, gender, clinical history). 

Response: Thank you for this valuable suggestion. We summarized the age and gender of the 28 cases and added appropriate references.

Kiyama et al have previously reported 15 cases of congenital methemoglobinemia [18] and found that congenital methemoglobinemia is common on the islands of Henza and Miyagi in Okinawa, Japan. They surveyed 2,876 islanders on the islands of Henza and Miyagi and found an additional 13 cases, leading to a total of 28 cases in 12 families. Of the 28 cases, 46.4% (13 of 28) were male; mean age was 27.3 years (median: 22, standard deviation: 15.8), 3 were over the age of 50 years, and 1 was 70-year-old; 25 were asymptomatic, 1 had epilepsy, and 2 had mental retardation.

  • Introduction: More background on how Cyb5R3 functions in vivo and how it affects red blood cells (for instance, reduction of ferric iron to ferrous iron to enable oxygen transport) would be useful to the reader.  How the Arg58Gln point mutation affects Cyb5R3 function, if known, should be included.

Response: Thank you for this valuable suggestion. Accordingly, we have added the relevant text and added a corroborating figure and references.

Methemoglobin is a form of the protein hemoglobin in which the iron is in the ferric state, rather than the normal ferrous state.Although a small percentage of methemoglobin is present in healthy individuals, an increase in methemoglobin content occurs due to loss of function of CYB5R3.

  • Reference 10:  incomplete citation. See https://lib.dmu.edu/db/uptodate/cite.

Response: Thank you for pointing this out. We have corrected this to a valid citation link.

Reviewer 2 Report

This is a rare case report presenting as hereditary congenital methemoglobinemia .  My comment as following

Please summary the clinical characteristics of reported cases (references 3-7) and current case. 

Author Response

Please summary the clinical characteristics of reported cases (references 3-7) and current case. 

Response: Thank you for your encouraging comments and valuable suggestions.

Many cases are diagnosed with congenital methemoglobinemia on the basis of underlying cyanosis.

We have added this information to the Conclusion section of the revised manuscript.

Reviewer 3 Report

Dear Editors and Reviewers,

I have read the paper "Hereditary Congenital Methemoglobinemia Diagnosed at the 2 Age of 79 Years: A Case Report" with a great interest.

The structure of the paper is accurate.  Authors present a case about patients with hereditary congenital methemoglobinemia diagnosed at the 2 age of 79 years and provide an interesting, comprehensive discussion on the topic.

However, I have some suggestions regarding this paper.

1. Conlclusions in abstract and in the end of the text should be modified. The information that only 4 studies have been published is not the conlcusion, but rather a result of searching the literature described earlier.

Similarly, sentence "We present a case of congenital methemoglobinemia first diagnosed at the age 184 of 79 years" is not a conlcusion.

2. Page, line 33: Please correct the sentence (there is lack "is").

Author Response

Response: Thank you for your encouraging comments and feedback.

First, we revised the Conclusion in the Abstract based on your comment. Second, we corrected the error in the sentence as you pointed out (add “is”).